# Construction of Landscape Ecological Security Pattern in the Zhundong Region, Xinjiang, NW China

**DOI:** 10.3390/ijerph19106301

**Published:** 2022-05-22

**Authors:** Jiao Jiang, Abudukeyimu Abulizi, Abdugheni Abliz, Abudoukeremujiang Zayiti, Adila Akbar, Bin Ou

**Affiliations:** 1College of Geography and Remote Sensing Sciences, Xinjiang University, Urumqi 830046, China; jiangjiao@stu.xju.edu.cn (J.J.); abdugheni.abliz@xju.edu.cn (A.A.); adla13105893915@163.com (A.A.); ob1094367032@163.com (B.O.); 2Xinjiang Key Laboratory of Oasis Ecology, Xinjiang University, Urumqi 830046, China; 3MNR Technology Innovation Center for Central Asia Geo-Information Exploitation and Utilization, Urumqi 830046, China; 4College of Ecology and Environment, Xinjiang University, Urumqi 830046, China; krimjan411@xju.edu.cn

**Keywords:** Zhundong region, MSPA method, MCR model, landscape ecological security pattern

## Abstract

The Xinjiang Zhundong Economic and Technological Development Zone, which contains the largest coalfield in China, is a mega energy base for west–east gas transmission and outbound electricity transmission in China; however, resource exploitation and the region’s arid climate have led to the region’s ecological environment being increasingly vulnerable. The morphological spatial pattern analysis (MSPA) method and landscape connectivity were used in this study to identify the ecological sources and extract the ecological corridors and ecological nodes based on the minimum cumulative resistance (MCR) model, used to construct the landscape ecological security pattern in the Zhundong region from 2016 to 2021. The results show that (a) from 2016 to 2021, the area of ecological sources increased by 117.86 ha and the distribution density of which decreased from the southern-central region to the northern and northwestern regions. (b) From 2016 to 2021, the number of ecological corridors and ecological nodes decreased, and the ecological corridors with dense distributions in the south gradually moved to the north and west. The length of the ecological corridors in the south gradually became longer, and the number of ecological corridors connecting the east and west in the north increased. (c) The landscape ecological security pattern of the Zhundong region was constructed by “a network and multiple points” using the model of ecological sources–ecological corridors–ecological nodes. The findings of this study provide a scientific foundation for the construction of an ecological security development plan and the ecologically protective development of coal resources in Zhundong.

## 1. Introduction

The Xinjiang Zhundong Economic and Technological Development Zone (hereinafter referred to as the Zhundong region) is a large-scale coal, electricity, and chemical industry demonstration zone founded on the planning and construction of the Zhundong coalfield. The Zhundong coalfield is an essential part of China’s 14th large-scale coal base. Not only does this area contain the largest packaged coalfield in China at present, but it is also home to a national geopark, wildlife and grassland nature reserve, and national desert nature parks. However, due to its location in the Gobi Desert region, precipitation is scarce, the climate is arid, and vegetation is sparse, resulting in its originally arid and fragile ecological environment being constantly superimposed on by the socioeconomic activities related to the large-scale coal mining [1,2]. The regional ecological landscape has changed dramatically, and the pressure for ecological environmental protection has intensified, threatening green coal mining and ecological security in Zhundong. Thus, it is imperative to strengthen the ecological security protection in the study area [3,4].

The landscape ecological safety pattern is a prospective spatial pattern composed of certain significant localities, locations, and spatial connections in the landscape [5], and it is an important component of China’s territorial spatial planning strategy [6]. The effective construction and maintenance of regional landscape ecological security patterns is beneficial to the ecosystem structure and functions, biodiversity conservation, and maintenance of ecosystem services, as well as to improving the well-being of humans, achieving regional sustainable development, and ultimately ensuring ecological security [7,8]. Constructing an ecological security pattern is regarded as a critical means of systematically solving ecological security issues and promoting high-quality sustainable development [9,10]. Therefore, how to construct a landscape ecological security pattern for an ecologically fragile area and to coordinate the relationship between the environmental conservation and socioeconomic development has become a much-debated topic in current research [11,12].

In the 1990s, Forman [13] systematically summarized the current landscape pattern optimization approaches, and Yu [14] introduced the concept of the landscape ecological security pattern and enhanced the minimum cumulative resistance (MCR) model, which has been widely applied to constructing landscape ecological security patterns. Horvath [15] evaluated the impact of open-pit mining on ecological security patterns as early as the end of the 20th century. At present, the research framework of identifying sources, constructing the resistance surface, and extracting corridors has become the fundamental paradigm for constructing regional landscape ecological security patterns [16,17]. Wu et al. [18] constructed a landscape ecological security pattern using the multi-regulation integrated resistance surface model, which was beneficial to the ecological restoration and landscape ecological management of grassland mining cities in semi-arid regions. Peng [19] and Zhang et al. [20] constructed a local ecological security pattern using the minimum cumulative resistance model for an arid area with a fragile ecological background in the north. Huang Xin et al. [21] used the MCR model to construct a landscape ecological security pattern for the Shengli coalfield in Inner Mongolia and provided recommendations for the protection of the ecological sources, ecological corridors, and ecological nodes. Zhao Liangfei et al. [12] considered the fragile ecological environment, functional positioning, and construction objectives of the Shendong mining area. Using the MCR model, they constructed the landscape ecological security pattern for the Shendong mining area. Li Hengkai et al. [22] used the MCR model approach to analyze the landscape ecological security pattern in the southern rare earth mining areas. They proposed an optimization strategy for the local ecological security pattern of the rare earth mining area in order to provide scientific decisions for managing the ecological environment of the mining area. Zhao et al. [23] examined the geographical and temporal characteristics of the ecological security pattern in an environmentally vulnerable area from 1990 to 2019 using the MCR model, identified the core ecological security areas, and predicted the ecological pattern in the Kalamaili region in 2025. Although some research has been undertaken on the ecological security assessment of the Zhundong open-pit coal mine in Xinjiang [24], few studies have been conducted on the construction of landscape ecological security patterns for the Zhundong region. There is an urgent need to reconcile the ecological security and resource development in ecologically fragile areas. Therefore, this paper aims to (1) identify ecological sources based on the MSPA method and landscape connectivity; (2) construct a landscape ecological security pattern of the region based on the MCR model; and (3) optimize the landscape ecological security pattern. Moreover, this ecological security pattern can serve as a scientific foundation for implementing ecological conservation and a reference for constructing landscape ecological security patterns and developing ecological security in arid open-pit coal mining areas of the world.

In this study, the Zhundong region was taken as the research object, the MSPA method and landscape connectivity index were used to select the ecological sources, and the MCR model was used to identify the ecological corridors and ecological nodes, resulting in the landscape ecological security pattern of the Zhundong region. The results provide suggestions for ecological conservation and developing ecological security in arid open-pit coal mining areas of the world.

## 2. Materials and Methods

### 2.1. Study Region

The Xinjiang Zhundong Economic and Technological Development Zone is mainly located in the Changji Hui Autonomous Prefecture of the Xinjiang Uygur Autonomous Region, at the northern foot of the Tianshan Mountains and on the southern margin of the Junggar Basin (44°6′–45°14′ N, 88°43′–90°59′ E), spanning Jimsar County, Qitai County, and Mubi Kazakh Autonomous County, with a total area of about 1,550,000 ha (Figure 1). The Zhundong coalfield in the study area has five mining areas, which are under the jurisdictions of Wucaiwan, Dajing, Xiheishan, Jiangjunmiao, and Laojunmiao. It is one of the five major coalfields in Xinjiang, with total predicted coal resources of 390 billion tons and 253.1 billion tons of proven reserves, making it the largest packaged coalfield in China at present [25]. The study area is also home to the Xinjiang Qitai Silicified Wood Dinosaur National Geological Park, the Xinjiang Kalamaili Mountain Ungulate Nature Reserve, the Qitai Desert and Grassland Nature Reserve, the Xinjiang Jimsar National Desert Nature Park, and the Xinjiang Mulei Mingsha Mountain National Desert Nature Park. It has a temperate continental climate with an extreme maximum temperature and an extreme minimum temperature of 41.6 °C and −36.6 °C, respectively. The average annual precipitation is 191 mm, the annual evaporation is 2046 mm, the vegetation is sparse, and the ecological environment is extremely fragile [26]. Therefore, preserving the ecological safety and structuring the ecological pattern of the landscape in the study region is a critical scientific issue that urgently needs to be addressed.

### 2.2. Data Sources and Preprocessing

#### 2.2.1. Data Sources

In this study, Sentinel 2 remote sensing images provided by the European Space Agency (ESA) (https://scihub.copernicus.eu/ (accessed on 29 November 2021)), with a spatial resolution of 10 m, were used to effectively monitor the vegetation information. According to previous research results [25,27] and fieldwork, the vegetation in the study area grows best in May and June, followed by July and August. As a result, the images of the study area with less cloud cover and greater vegetation growth were chosen. A digital elevation model (DEM) with a spatial resolution of 30 m was obtained from the Geospatial Data Cloud (http://www.gscloud.cn/ (accessed on 19 November 2021)) and was used to extract the slope and height data. The vector map boundary of the study area was derived from the spatial planning of the national land in the Zhundong Economic and Technological Development Zone (2020–2035). The road data for the Zhundong region were obtained from Open Street Map (https://www.openstreetmap.org (accessed on 18 December 2021)).

#### 2.2.2. Data Preprocessing

Due to inconsistencies in the Sentinel 2 data products for 2016 and 2021 (Table 1), radiometric calibrations and atmospheric corrections are required for the 2016 Level 1C products in the Sen2Cor. The band synthesis and resampling of remote sensing images in SNAP were performed to generate images with 10 m resolution.

ENVI5.3 was used to classify the remote sensing images of the research area in 2016 and 2021 using the support vector machine method. According to the research needs, the land use types were classified into a total of six categories: water bodies, vegetation, construction land, coal mine land, unused land, and other types of land (Figure 2). Based on high-resolution Google remote sensing images for the same year and a field survey conducted for accuracy verification, the interpretation accuracy was greater than 80%, which fully meets the needs of this study of the Zhundong region.

### 2.3. Methods

#### 2.3.1. Classification of Land Use

The supervised classification method is the predominant approach in land cover identification, with remarkable performance in remote sensing image classification [28]. In this paper, the support vector machine (SVM) method was used to classify the land use of remote sensing images in the Zhundong region.

#### 2.3.2. Calculation of NDVI

The normalized difference vegetation index (NDVI) was used to monitor vegetation growth status, vegetation cover and to eliminate some of the radiometric errors [28].
(1)$$\mathrm{NDVI}=\frac{\rho_{NIR}-\rho_{Red}}{\rho_{NIR}+\rho_{Red}}$$
when −1 ≤ NDVI ≤ 1, a negative value indicates that the ground is covered with clouds, water, or snow. A value of 0 indicates rock or bare soil, etc. A positive value indicates that there is vegetation cover and the value increases with increasing coverage.

#### 2.3.3. Identification of Ecological Sources Based on MSPA Method

The MSPA method is a spatial pattern analysis method that can be used to determine the types and structures of the landscape more precisely. It was developed by Vogt [29] and other scholars based on mathematical morphological principles such as erosion, expansion, open operation, and closed operation to measure, identify, and segment the spatial patterns of raster images [30,31,32].

First, the water bodies and vegetation land use types in the study area were used as the foreground for the MSPA, and the remainder of the land use types were utilized as the background based on the results of the remote sensing image interpretation in the Zhundong region in 2016 and 2021. Second, Guidos 2.6 software (the European Commission, Joint Research Centre (JRC), Ispra, Italy) was used to analyze the data using the eight-neighborhood analysis method in the MSPA. Seven categories of the seven landscape types, including core, branch, edge, islet, bridge, loop, and perforation, were identified in the study area in 2016 and 2021 [33]. Finally, the results of this analysis were tallied, and the top ten core areas of habitat patches were extracted as the ecological sources [34].

#### 2.3.4. Selection of Ecological Sources Based on Landscape Connectivity Index

The landscape connectivity reflects the degree to which the landscape facilitates or impedes ecological flow, and maintaining good connectivity is key to promoting biodiversity and sustaining ecosystem stability and integrity [31,35,36]. Based on the goals of this study, the overall connectivity index (IIC) and the delta of PC (dPC) were selected to analyze the landscape connectivity and patch importance of the core areas. The core areas with higher IIC and dPC values were chosen to identify the ecological sources in the Zhundong region in order to better evaluate the connectivity among the core patches in the region.

Using the Conefor software (Polytechnic University of Madrid (Madrid, Spain) and University of Lleida (Lleida, Spain)), the patch connectivity threshold was set to 500 m, and the connectivity probability was set to 0.5 based on previous research results [37,38,39,40] and the actual conditions of the study region. The 10 patches with the most significant dPC values were selected as the ecological sources to assess the landscape connectivity of the Zhundong region. The formulas used are as follows.

Integral index of connectivity (IIC):(2)$$\mathrm{IIC}=\frac{\sum_{i=1}^{n}\sum_{j}^{n}\frac{a_{i}\cdot a_{j}}{1+nl_{ij}}}{A_{L}^{2}}$$

Delta of PC (dPC):(3)$$\mathrm{dPC}=100\frac{PC-PC_{remove}}{PC}$$

In Equations (2) and (3), *n* is the number of patches in the research region.$a_{i}$ and $a_{j}$ are the areas of patches *i* and *j*, respectively; $A_{L}$ is the total area of the landscape in the study area; *PC* indicates the degree of possible connectivity of a given patch in the study area; and $PC_{remove}$ is the value of the landscape connectivity index if the patches are removed from the study area [41].

#### 2.3.5. Construction of Resistance Surface

The MCR model refers to the cost required for the migration movement of species from ecological sources to destinations, and it was first proposed by Knaapen et al. [42] and was then modified by Yu Kongjian et al. They combined it with the cost distance problem in a geographic information system (GIS) to identify the key areas and key nodes of the landscape ecology, and it has become the primary tool for creating landscape ecological security patterns [43]. It is calculated as follows:(4)$$\mathrm{MCR}=fmin\overset{i=m}{\underset{j=n}{\sum }}\left(D_{ij}R_{i}\right)$$
where MCR is the minimum cumulative resistance of ecological source *i* to any point *j*. $D_{ij}$ is the spatial distance from species *j* to landscape unit *i*; and $R_{i}\;$ is the resistance of landscape unit *i* to the migration of a certain species [44].

The land use type, NDVI, slope, DEM, distance from roads, and distance from coal mine sites were chosen as the resistance factors in this study based on the principles of selectability and quantifiability, as well as the actual situation in the study area, which took into account the natural, human, and socioeconomic factors. In the calculation of ecological resistance surface, the relationship between the five indicators and resistance value is as follows: NDVI is inversely related to resistance value and slope is positively related to resistance value. This paper mainly considers the influence of human activity intensity. Land use classification is different, and human activity intensity is different. Therefore, land use classification is divided into different grades. The relationship between the hierarchy of resistance is as follows: the greater the anthropogenic disturbance, the greater the intensity of land use, and the greater the resistance to species movement. The influence of the distance from roads and from coal mines are reflected by the distance, the further the distance, the lower the resistance; the closer the distance, the higher the resistance. The smaller the resistance value, the less resistance the species’ spatial dispersal process has to overcame. According to the relevant research results [9,12,37], the relative resistance values ranged from 1 to 500. The factors of each type are divided into six levels for assignment and reclassification, and then the resistance surface is calculated by multiplying the weight by the resistance value.

The weight coefficients in this study were determined by the analytic hierarchy process (AHP). The importance scales of different indicators in this method and their meanings are shown in Table 2 [45].

Combined with the actual situation of the Zhundong region, the area is located in arid ecologically fragile area, the more the disturbance by human beings, the more the impact on its ecological environment. Therefore, land use has the highest influence, followed by distance from the coal mine and the road. Compared with slope, DEM and NDVI, these factors have a greater impact on ecological environment disturbance. The slope and DEM have a greater impact on the species habitat than NDVI (Table 3).

#### 2.3.6. Construction of Ecological Corridors

Ecological corridors are the optimal pathways for the maintenance and dispersal of different species and are the channels of material and energy flow in natural ecosystems [46]. They can communicate the type of ecosystem space that connects the ecological units that are more isolated and dispersed in the spatial distribution [2]. In this study, we used the cost distance tool in ArcGIS to obtain the minimum cumulative distance surface in the Zhundong region and then we used the cost path to extract the ecological corridors. Finally, we eliminated the redundant part to obtain the ecological corridors in the study area based on the minimum cumulative resistance model, the results of the ecological source identification, and the integrated resistance surface.

#### 2.3.7. Identification of Ecological Nodes

Ecological nodes are generally distributed in areas with the weakest ecological functions, which are located in the ecological corridors that connect the ecological sources, and they play a vital role in the regional ecological flow [47]. In this study, the ridgelines were identified using the hydrological analysis tool in ArcGIS, and the ecological corridors and their intersections were defined as the ecological nodes in the study area [21].

## 3. Results

### 3.1. Identification of Ecological Sources Using MSPA

As shown in Table 4 and Figure 3, the ecological land in Zhundong was generally sparsely distributed. The ecological land was largely distributed in the southern-central part of the study area, with a tiny proportion in the west and north. The remaining portion was scattered across the study area. Moreover, the patches in the core area had a long-distance and poor connectivity, which hindered the exchange of materials and energy between biological species. Compared with 2016, the core area of patches in the study area decreased by 458.89 ha in 2021, which was mainly due to the reduction of the fragmented patches in the southern-central part of the study area. The edge area decreased by 712.78 ha, indicating that the edge effect became worse and these areas became more vulnerable to external disturbance from 2016 to 2021. The islet component was fragmented, and the area decreased by 436.90 ha. The perforation component remained almost unchanged, and its percentage was small. The reduction of the branch weakened the bioenergy exchange and material flow in the study area. The reduction of the bridge and loop components indicates that the migration channels between the organisms and within the patches decreased.

### 3.2. Analysis of Ecological Source Extraction Based on Landscape Connectivity

Based on the data presented in Table 5 and Table 6, the top ten ecological sources in terms of area were chosen within the Zhundong region in accordance with the delta of PC (dPC) values. Overall, the total area of the ecological sources in the study area increased by 117.86 ha and the distribution of the ecological sources was uneven. The core patches in the Huangcao Lake area in the southern-central section of the study area (Figure 3) had the highest degree of connectivity. This indicates that the habitat patches in the southern-central area were more appropriate for the migration of biological species and the transfer of material and energy and could also better provide habitats and sources for species. The dIIC and dPC values of the various ecological sources were quite diverse, and the connectivity between the patches was weak. Compared with 2016, the core with the largest area in 2021 was still located near the Huangcao Lake area. In addition, the patch area increased by 183.01 ha and the connectivity increased by 13.49. The ecological source sites migrated from the concentrated southern area to the northwestern area near the new city of Zhundong, and the area of the ecological sources in the northwest was 70.24 ha. In 2021, the increased ecological sources were mainly located in the Qitai Desert Grassland Nature Reserve and the Wucaiwan area, while the ecological sources near Beitashan Mountain and the Qitai County Sand Control Station decreased. From 2016 to 2021, the number of patches with dPC values of greater than 1 decreased and the dPC value decreased continuously, which indicates that the degree of fragmentation of the patches increased.

### 3.3. Resistance Surface

The resistance factors were divided into different levels and assigned values, and the weights were calculated using the AHP. To obtain the comprehensive resistance surface, the resistance system was built and the single spatial resistance factor surfaces were overlain (Figure 4).

As can be seen from Table 7, the land use type was the most important resistance factor in the Zhundong region, followed by the distance to roads and the distance to coal mine sites, and finally the slope, DEM, and NDVI had less influence. After assigning each resistance value, the consistency test result was CR = 0.0816 < 0.1, which passed the one-time test. The resistance values were 70.7114–429.14 and 70.7114–435.18 in 2016 and 2021, respectively (Figure 4). The maximum resistance value in 2016–2021 increased by 6.04, while the minimum resistance value remained unchanged. The main reason for this was that the coal mine land and construction land near Wucaiwan in the northwest and Beitashan in the northeast increased significantly from 2016 to 2021, increasing the resistance value in the study area, which was not conducive to species migration. The changes in the vegetation and water bodies were not significant, so the minimum resistance value was almost constant. The areas with high resistance values were primarily located on roads, integrated service facility bases, and in mining areas where human activities were frequent and had a significant impact on and damaged the regional ecological environment. The areas with low resistance values were mainly located in water bodies, wetlands, and areas with vegetation cover. The ecological environment in these areas was relatively favorable, providing supplies for species migration. The resistance values of these areas were low because of the unique local environmental conditions, including some bare land and sparse vegetation which can provide habitats for species.

### 3.4. Landscape Ecological Security Pattern

#### 3.4.1. Ecological Corridors

Based on the MSPA method and MCR model, in this study, 38 ecological corridors were identified in the Zhundong region in 2016, with a length of 1543.33 km, which were densely distributed in the southern-central part of the study area and sparsely distributed in the northern corridors. Due to the presence of more vegetation in the south, the distance between ecological sources was relatively small, with less landscape resistance and better connectivity, facilitating species migration. There was much unused land in the north, and the length of the corridors connecting the sources was long, with high landscape resistance. There were 36 ecological corridors in the study area in 2021, with a length of 1328.78 km, and they were mainly distributed in the central part of the study area, with a long overall corridor length. However, the northwestern ecological corridors were shorter in length, with less landscape resistance and better connectivity. Compared with 2016, the total number of ecological corridors decreased, and the overall distribution was relatively uniform in 2021, but the number of ecological corridors in Wucaiwan increased. The main reason for this was the increase in the living service facilities in the vicinity of Wucaiwan, the construction of urban green parks, and the increase in ecological sources, which increased the ecological corridors. As the government strengthened the protection of nature reserves and natural wilderness areas, the number of ecological corridors from east to west in the northern part of the study increased. The decrease in the number of ecological corridors in the northeast was due to the increase in the number of coal mines and industrial facilities and the amount of construction land in the Beitashan area, which cut off these corridors.

#### 3.4.2. Ecological Nodes

The hydrological analysis tool was used to extract the study area’s ridgelines in 2016, and 28 ecological nodes were identified in the study area. The ecological nodes were mainly densely distributed in the southern part of Lake Jiji and in Xiheishan, while the ecological nodes in the western part were sparsely distributed. There were 20 ecological nodes in 2021, which were mainly distributed in the vicinity of Jiangjunmiao and Xiheishan in the east, the nature reserves and nature parks in the north, and a few in Dajing in the west. The overall distribution of the ecological nodes was scattered, which was greatly influenced by the roads. Compared with 2016, although the overall number of ecological nodes decreased in 2021, they were no longer densely distributed near Lake Jiji. However, they gradually moved farther to the north near the Xinjiang Kalamaili Mountain Ungulate Nature Reserve and the Qitai Desert and Grassland Nature Reserve and became more scattered. The number of ecological nodes near Wucaiwan in the northwest also increased. The Zhundong Economic and Technological Development Zone Territorial Spatial Plan (2012–2030) required adherence to ecological priority, strict protection of ecological land, and the use of facility corridors, road protection green belts, and ecological green areas between groups, combined with the current ecological substrate, to construct a regional ecological network and achieve sustainable development. Therefore, the ecological nodes gradually became dispersed around the ecological reserves and the ecological green areas in the study area from 2016 to 2021. The construction of road buffer zones and green spaces caused the ecological nodes to be distributed along the roads.

## 4. Discussion

### 4.1. Landscape Pattern Characteristics

The landscape composition of the Zhundong region was relatively simple, with vast areas of bare land and sparse vegetation growth, accounting for more than 95% of the total area of the study area. With the development of resources and the economy, the amount of construction land has increased. However, the ecological environment is fragile. There are natural reserves and national parks in the study area, so protecting the landscape’s ecological safety is essential in Zhundong. The vigorous mining of coal and the increase in construction land led to an increase in the number of sporadic ecological patches and a decrease in the area of the ecological patches, thus increasing the fragmentation of the landscape and reducing the connectivity of the regional landscape.

### 4.2. Ecological Sources Optimization

Due to the fragile ecological environment in arid areas, it is hard for nature to form an ecological pattern, owing to the disturbance by the coal industry [32]. Therefore, the ecological security of the Zhundong region should be preserved in accordance with the requirements of ecological civilization. The nature reserves, historical and cultural nodes, and national geological parks in the study area should be protected, and their biodiversity should be preserved. In the water-scarce areas in Zhundong, it is vital to protect the water sources, clarify the scope of the water source protection, and prohibit construction activities within the water source protection area [48]. The ecological patch with the highest connectivity remained in the Huangcao Lake area in the south, which is suitable for species’ habitats and migration. With poor regional circulation between the east and west in the northern part of the study area, it is critical to improve the protection and construction of ecological sources suited for species’ habitats and migration in the northeast and northwest to promote sustainable ecosystem development [46].

### 4.3. Ecological Security Pattern Construction Optimization

The ecological security pattern constructed in this study is consistent with the “Six axes and multiple points” proposed in the Territorial Spatial Plan for the Zhundong Economic and Technological Development Zone (2021–2035). Due to the fragile ecological environment, single plant community, and sparse vegetation growth in the study area, it is necessary to adhere to the ecological red line, protect the biodiversity in Zhundong, maintain the stability of the ecological system, and construct the landscape ecological security pattern of the region. The ecological sources within the study area can be increased, and measures such as buffer zones can be implemented around the ecological sources [12]. Ecological green parks and other facilities can be built around residential areas, the amount of vegetation should be increased in the mining development areas, and coal mining rehabilitation and revegetation measures should be carried out. Due to the lack of ecological corridors connecting the east and west in the central part of the region, native vegetation should be selected for ecological restoration when constructing the central east–west ecological corridors. Vegetation can also be added on both sides of the roads to increase the connectivity between the east and west. In the ecological corridors and ecological nodes, buffer zones and artificial vegetation planting should be implemented along the roads and in the construction land and coal mine land, the connectivity of the biological flow and migration should be increased, and the stability of the ecological system should be maintained. Green space facilities should be constructed in the comprehensive living service bases in the Wucaiwan and Lake Jiji areas and in service points such as Huoshaoshan and Jiangjunmiao.

## 5. Conclusions

In this study, the MSPA method and landscape connectivity were used to identify and select the ecological sources in 2016 and 2021 in the Zhundong region. According to resistance factors such as the NDVI, land use, DEM, distance to roads and mines, the resistance surface in Zhundong was constructed. The ecological corridors and ecological nodes were extracted using the MCR model and the resistance surface constructed in the study area, resulting in the landscape ecological security pattern of the Zhundong region. The conclusions of this study are as follows.

(1)From 2016 to 2021, the ecological source area in the Zhundong region increased by 117.86 ha, and the connectivity also increased. The main land-use types of the ecological sources were water bodies and vegetation, which gradually changed from a dense distribution in the Huangcao Lake area and near Lake Jiji in the south to the nature reserves in the north and Wucaiwan New Town in the northwest.(2)From 2016 to 2021, the numbers of ecological corridors and ecological nodes decreased, and the ecological corridors with a dense distribution in the south gradually moved to the north and west. The length of the ecological corridors in the south steadily increased, and the number of ecological corridors connecting the east and west in the north increased. Most of the ecological nodes were located near construction land and roads, which had relatively fragile ecological functions, and they gradually spread from the vicinity of Lake Jiji in the south to Wucaiwan in the northwest and the nature reserves in the north.(3)In this study, the landscape ecological security pattern of Zhundong, including the ecological sources, ecological corridors, and ecological nodes, was constructed using the MSPA method and the MCR model. The results of this study lay a foundation for optimizing the landscape ecological security pattern in the study area and provides a scientific basis for the protective development and ecological management of local coal resources.

The identification of ecological sources is a central part of the ecological security pattern construction process. In this study, the MSPA method and landscape connectivity were used to identify the ecological sources; this is a more scientific method and effectively avoids the high error resulting from manual selection of sources. To a certain extent, it makes up for the deficiencies of existing studies. The identification and analysis of the ecological corridors and ecological nodes was effectively used to construct a landscape ecological security pattern, enriching the research on constructing landscape ecological security patterns in arid, ecologically fragile areas, maintaining regional environmental protection, and promoting collaborative economic development. However, there is some subjectivity in the setting of the connectivity threshold and the relative resistance value in this study, and further refinement of the deficiencies of the study is needed in the future.

## Figures and Tables

**Figure 1 ijerph-19-06301-f001:**
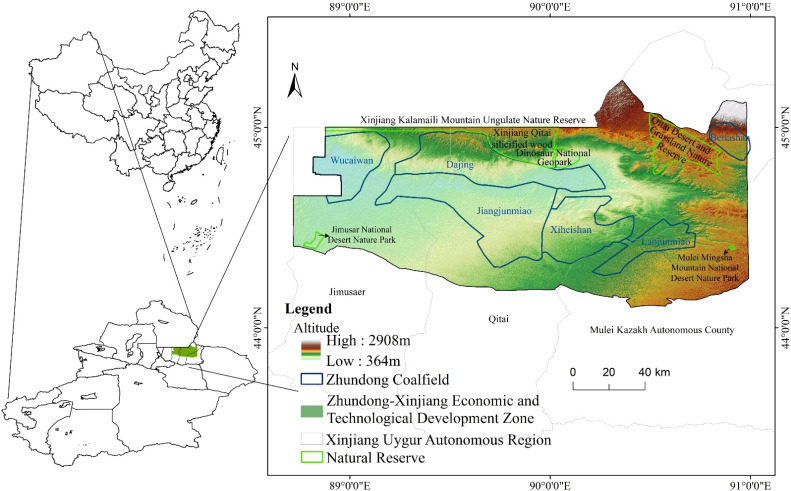
Geographic location of the study area.

**Figure 2 ijerph-19-06301-f002:**
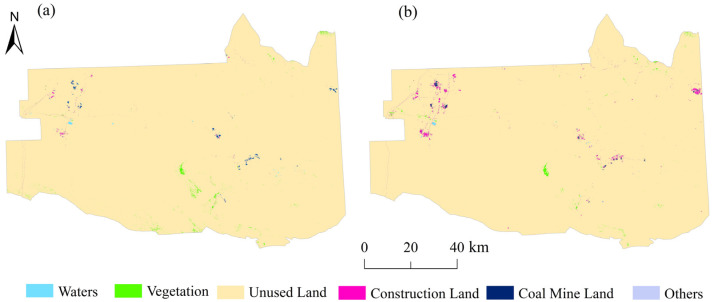
Land use type maps for (**a**) 2016 and (**b**) 2021.

**Figure 3 ijerph-19-06301-f003:**
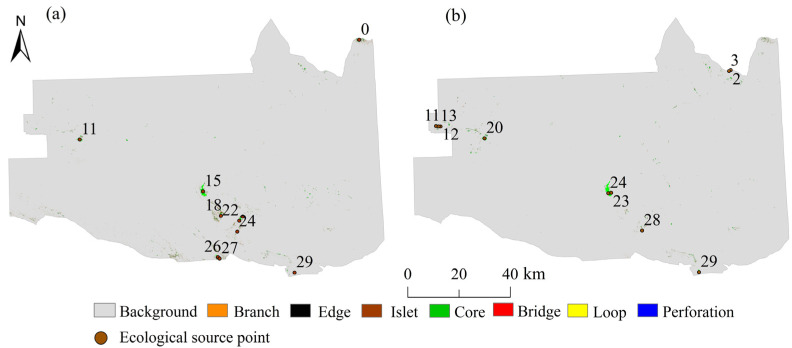
Identification of ecological sources in (**a**) 2016 and (**b**) 2021.

**Figure 4 ijerph-19-06301-f004:**
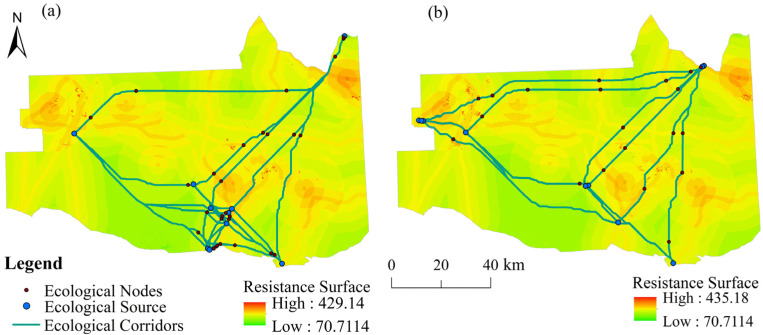
Landscape ecological security patterns for (**a**) 2016 and (**b**) 2021.

**Table 1 ijerph-19-06301-t001:** Remote sensing image data of Sentinel 2 in study area.

Year	Generation Date	Product Type	Relative Orbit	Tile Identifier
2016	05-23	S2MSI1C	76	46TCQ
	07-02		76	T45TYJ
	07-15		119	T45TYK
	07-15		119	T45TYL
	07-15		119	T45TXK
2021	07-21	S2MSI2A	76	46TCQ
	07-21		76	T46TCR
	07-24		119	T45TXK
	07-24		119	T45TYK
	07-24		119	T45TYJ
	07-26		76	T46TCP
	08-03		119	T45TYL

**Table 2 ijerph-19-06301-t002:** Indicator importance scale.

Importance Scale *a_ij_*	Description
1	Two factors have the same importance
3	*i* is slightly more important than *j*
5	*i* is more important than *j*
7	*i* is much more important than *j*
9	*i* is extremely more important than *j*
2, 4, 6, 8	scale median

**Table 3 ijerph-19-06301-t003:** Analytic hierarchy process judgment matrix.

Indicators	Land Use Classification	NDVI	Slope	DEM	Distance to Road	Distance to Coal Mine
Land use classification	1	5	7	7	3	3
NDVI	1/5	1	1/3	1/3	1/3	1/3
Slope	1/7	3	1	1	1/5	1/5
DEM	1/7	3	1	1	1/5	1/5
Distance to road	1/3	3	5	5	1	1
Distance to coal mine	1/3	3	5	5	1	1

**Table 4 ijerph-19-06301-t004:** Statistical results of MSPA classification from 2016 to 2021.

Landscape Type/Year	Total Area (ha)		Percentage of the Ecological Land (%)	
	2016	2021	2016	2021
core	2625.09	2166.20	45.30	59.22
Islet	766.79	329.88	13.23	9.02
Perforation	36.22	32.88	0.63	0.90
Edge	1523.57	810.79	26.29	22.17
Loop	74.68	26.41	1.29	0.72
Bridge	177.49	59.86	3.06	1.64
Branch	591.12	231.93	10.20	6.34

**Table 5 ijerph-19-06301-t005:** Ranking of patch importance and landscape connectivity of ecological sources in 2016.

Patch Number	dIIC	dPC	Area (ha)
15	72.03	71.25	463.58
11	19.78	19.56	242.90
22	2.00	2.14	67.6
26	1.07	1.33	48.66
24	1.17	1.16	59.14
18	0.69	0.91	36.97
20	0.57	0.76	11.07
0	0.34	0.66	14.03
27	0.37	0.64	17.03
29	0.59	0.58	41.97

**Table 6 ijerph-19-06301-t006:** Ranking of patch importance and landscape connectivity of ecological sources in 2021.

Patch Number	dIIC	dPC	Area (ha)
15	72.03	71.25	463.58
11	19.78	19.56	242.90
22	2.00	2.14	67.6
26	1.07	1.33	48.66
24	1.17	1.16	59.14
18	0.69	0.91	36.97
20	0.57	0.76	11.07
0	0.34	0.66	14.03
27	0.37	0.64	17.03
29	0.59	0.58	41.97

**Table 7 ijerph-19-06301-t007:** Resistance values and weights for the study area.

Resistance Factor	Classification	Resistance Value	Weight
Land use classification	vegetation	1	
	water bodies	50	
	unused land	200	0.4289
	construction land	400	
	coal mine land	500	
NDVI	[−1, 0)	100	0.0437
	[0, 0.2)	500	
	[0.2, 0.4)	350	
	[0.4, 0.6)	250	
	[0.6, 0.8)	125	
	[0.8, 1)	1	
Slope (°)	[0, 5)	100	0.0604
	[5, 15)	200	
	[15, 25)	300	
	[25, 35)	400	
	[35, 90)	500	
DEM (m)	[0, 300)	100	0.0604
	[300, 400)	200	
	[400, 500)	300	
	[500, 800)	400	
	[800, ∞)	500	
Distance to road/coal mine (m)	[0, 200)	500	0.2033
	[200, 400)	400	
	[400, 600)	300	
	[600, 800)	200	
	[800, ∞)	100	

## Data Availability

Data is contained within the article.

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
