# Peer review of "Construction of Landscape Ecological Security Pattern in the Zhundong Region, Xinjiang, NW China"

_ijerph, 2022, doi:10.3390/ijerph19106301_

Round 1

Reviewer 2 Report

This study discusses construction of landscape ecological security patterns in the study area using  morphological spatial pattern analysis, landscape connectivity, and minimum cumulative resistance (MCR) model. The authors discussed the methods used and presented the subsequent findings. I have two questions:

  1. How did the authors calculate resistance factors? More detailed discussion regarding this is needed.
  2. Similarly, I did not find details about how AHP was conducted to assign weights. Who were selected to do the pairwise comparison and why? These need to be discussed in detail considering how AHP is used in this study.

Round 2

Reviewer 1 Report

The authors addressed all of my comments and the paper was significantly improved.

This manuscript is a resubmission of an earlier submission. The following is a list of the peer review reports and author responses from that submission.